# Prognosis after Mild Traumatic Brain Injury: Influence of Psychiatric Disorders

**DOI:** 10.3390/brainsci10120916

**Published:** 2020-11-27

**Authors:** Ivan Marinkovic, Harri Isokuortti, Antti Huovinen, Daniela Trpeska Marinkovic, Kaisa Mäki, Taina Nybo, Antti Korvenoja, Raj Rahul, Risto Vataja, Susanna Melkas

**Affiliations:** 1Neurology, University of Helsinki and Helsinki University Hospital, Haartmaninkatu 4, Helsinki P.O. Box 340, FIN-00029 HUS Helsinki, Finland; harri.isokuortti@hus.fi (H.I.); antti.huovinen@helsinki.fi (A.H.); susanna.melkas@hus.fi (S.M.); 2Psychiatry, University of Helsinki and City of Helsinki, Nordenskiöldinkatu 20, P.O. Box 6800, FIN-00099 City of Helsinki, Finland; daniela.trpeskamarinkovic@hel.fi; 3Neuropsychology, University of Helsinki and Helsinki University Hospital, Haartmaninkatu 4, Helsinki P.O. Box 340, FIN-00029 HUS Helsinki, Finland; kaisa.maki@hus.fi (K.M.); taina.nybo@hus.fi (T.N.); 4HUS Medical Imaging Center, Radiology, University of Helsinki and Helsinki University Hospital, Haartmaninkatu 4, Helsinki P.O. Box 340, FIN-00029 HUS Helsinki, Finland; antti.korvenoja@hus.fi; 5Neurosurgery, University of Helsinki and Helsinki University Hospital, Topeliuksenkatu 5, Helsinki P.O. Box 266, FIN-00029 HUS Helsinki, Finland; raj.rahul@hus.fi; 6Psychiatry, University of Helsinki and Helsinki University Hospital, Välskärinkatu 12, Helsinki P.O. Box 590, FIN-00029 HUS Helsinki, Finland; risto.vataja@hus.fi

**Keywords:** brain concussion, return to work, mental disorders, post-concussion symptoms

## Abstract

Background: We evaluated the prevalence of psychiatric disorders in mild traumatic brain injury (MTBI) patients and investigated psychiatric comorbidity in relation to subjective symptoms and return to work (RTW). Methods: We recruited 103 MTBI patients (mean age 40.8 years, SD 3.1) prospectively from University Hospital. The patients were followed up for one year. The Rivermead Post-Concussion Symptom Questionnaire (RPQ) and Extended Glasgow Outcome Scale (GOSE) were administered one month after MTBI. Three months after MTBI, any psychiatric disorders were assessed using the Structured Clinical Interview for DSM-IV Axis I Disorders. Results: Psychiatric disorders were diagnosed in 26 patients (25.2%). The most common disorders were previous/current depression. At three months, there was no difference between patients with psychiatric disorders versus those without them in RTW (95.7% vs. 87.3%, *p* = 0.260) or at least in part-time work (100% vs. 94.4%, *p* = 0.245). In Kaplan–Meier analysis, the median time to RTW was 10 days for both groups. The median RPQ score was 13.0 (Interquartile range (IQR) 6.5–19.0) in patients with a psychiatric disorder compared to 8.5 (IQR 2.3–14.0) in those without one (*p* = 0.021); respectively, the median GOSE was 7.0 (IQR 7.0–8.0) compared to 8.0 (IQR 7.0–8.0, *p* = 0.003). Conclusions: Approximately every fourth patient with MTBI had a psychiatric disorder. These patients reported more symptoms, and their functional outcome measured with GOSE at one month after MTBI was worse. However, presence of any psychiatric disorder did not affect RTW. Early contact and adequate follow-up are important when supporting the patient’s return to work.

## 1. Introduction

Concussions and mild traumatic brain injuries (MTBI) are the most common forms of traumatic brain injury; recovery is rapid by default, but there are potential prolonged symptoms and adverse social, vocational and medicolegal issues [1]. It is challenging to identify individual factors that negatively influence recovery after MTBI [2].

Psychiatric symptoms and disorders are overrepresented in patients with MTBI both before and after the incident [3,4]. The most frequently reported disorders are mood disorders [5], anxiety disorders, psychotic disorders, post-traumatic stress disorder (PTSD) [6] and substance abuse-related disorders [7]. The psychiatric overall morbidity partly explains the markedly increased suicide rate in patients with MTBI [8,9]. Interestingly, MTBI patients have a tendency for prolonged psychiatric morbidity after a traumatic event compared to those suffering from moderate to severe TBI [6].

Return to Work (RTW) following TBI is a complex outcome parameter influenced by numerous factors, such as physical, cognitive, and emotional challenges [10]. Apart from cognitive intervention [11], RTW may be facilitated by other approaches, such as promoting a positive psychological perspective [12] and by various psychotherapeutic techniques [13,14]. These approaches, however, appear to be essential only for a small group of MTBI individuals (previously defined as the “miserable minority”) [15], who are characterized by delayed and complicated recovery. While MTBI clearly affects RTW [16], RTW after MTBI is mostly short [10]. However, the particular influence of psychiatric disorders in MTBI patients and their joint influence on RTW dynamics is so far understudied. On the other hand, adequate follow-up alone benefits RTW in MTBI patients. The aim of our study was to identify the presence of psychiatric disorders in patients with MTBI and their potential influence on reported symptoms and return-to-work dynamics. We hypothesized that the presence of psychiatric disorders might be related to more prominent symptoms and may cause a delay in return-to-work patterns.

## 2. Materials and Methods

### 2.1. Study Setting and Patients

The study was conducted at the Traumatic Brain Injury Outpatient Clinic of Helsinki University Hospital, which is the largest university hospital with a catchment area of approximately 2 million people. The TBI Outpatient Clinic serves as the only referral outpatient clinic for patients with MTBI in the metropolitan area. Referrals are accepted from all emergency departments (ED) in the area, both from city hospitals and from Helsinki University Hospital. General indications for the referral of MTBI patients are the following: head trauma that required acute CT scanning in ED, and the need for follow-up after dismissal from the ED or hospital ward.

In this prospective cohort study, we recruited 131 subjects with TBI, each of whom were residing in the Helsinki metropolitan area. Of the 131 patients, 103 underwent neuropsychiatric examination and were included in our study. A comparison between the excluded (*n* = 28) and the included patients (*n* = 103) revealed no significant differences (Table 1).

All patients were between 18 and 68 years of age, with Finnish language as a mother tongue. Exclusion criteria were the known presence of alcohol and/or drug addiction; previously diagnosed schizophrenia or schizoaffective disease; developmental, vision or hearing disability; and a contraindication for magnetic resonance imaging (MRI). In addition, patients were excluded if they had a known presence of alcohol and/or drug dependence revealed by patient records and by a structured interview before recruitment. However, alcohol abuse (not dependence) was not an exclusion criterion. The Diagnostic and Statistical Manual of Mental Disorders, 4th edition (DSM-IV) criteria were used to differentiate between dependence and abuse at the point of recruitment [17]. 

We used the World Health Organization (WHO) definition of MTBI [18]. These criteria include (1) one or more of the following: confusion or disorientation, loss of consciousness for 30 min or less, post-traumatic amnesia for less than 24 h, and/or other transient neurological abnormalities such as focal signs, seizure, and intracranial lesion not requiring surgery; and (2) a Glasgow Coma Scale score of 13–15 after 30 min post-injury or later upon presentation for healthcare. These manifestations of MTBI must not be due to drugs, alcohol, or medications, or caused by other injuries or treatment for other injuries (e.g., systemic injuries, facial injuries or intubation), other problems (e.g., psychological trauma, language barrier or coexisting medical conditions), or penetrating craniocerebral injury.

The recruitment process was accomplished by an experienced medical board—authorized attending neurologists at the TBI Outpatient Clinic of University Hospital during 2015–2018.

All included patients gave their written consent. The study complies with the provisions on data protection, international regulations governing the status of research patients, and medical research regulations. The study was approved by the local ethics committee (Helsinki University Hospital (HUH) 105/13/03/01/2014) (number 87, 09.04.2014).

### 2.2. Clinical Evaluation

Patients were evaluated by experienced specialists in neurology at the TBI Outpatient Clinic of University Hospital within one month after MTBI. The overall neurological condition was evaluated using the Neurological Outcome Scale for TBI (NOS-TBI), a measure developed to identify essential neurological deficits affecting the individuals with TBI [19]. Additionally, the Rivermead Post-Concussion Symptoms Questionnaire (RPQ) as self-report tool to assess the severity of post-concussion symptoms [20], and the Extended Glasgow Outcome Scale (GOSE) [21] as a global outcome measure in TBI survivors were utilized for analyses of clinical symptoms. Furthermore, the interview included questions about the current use of medications (antidepressants/benzodiazepines/other psychiatric medications). 

### 2.3. Brain Imaging

Most of the patients (96 out of 103) underwent brain CT scan in emergency, following a national consensus recommendation which recommends CT scan when certain criteria are filled. All imaging was performed with a 3T Siemens Magnetom Verio (Siemens, Erlangen) scanner with a 32-channel head coil. The imaging protocol consisted of a fast localizer, T1 sagittal localizer, axial FLAIR, coronal T2, 3D T2 SPACE, 3D T1 MPRAGE, 3D gradient-echo susceptibility-weighted imaging sequence, and a resting state BOLD FMRI (rs-FMRI) repeated twice with single-image volume with reversed phase encoding direction acquired after rs-FMRI time series for susceptibility-induced distortion correction. Gradient echo images for field mapping and distortion correction were acquired thereafter. In addition, diffusion-weighted images with 30 diffusion gradient (b = 1000) directions were acquired twice with reverse phase encoding directions (left-right/right-left). Four b = 0 image volumes were acquired with the two opposing phase encoding directions for susceptibility-induced distortion correction. To limit the total imaging time in the acute phase, the rs-FMRI time series was shorter (190 volumes, 5 min) than in the follow-up session (375 volumes, 9 min 45 s). In the follow-up session, multi-echo 3D gradient echo images were acquired as well. 

### 2.4. Psychiatric Evaluation

A psychiatric assessment was carried out 111 days (IQR 88–193), i.e., 3 months and 21 days, after incident MTBI. We used the Structured Clinical Interview for DSM-IV Axis I disorders (SCID I) tool to assess psychiatric disorders [22]. Each patient was examined using the structured questionnaire by a qualified examiner who has received SCID training. The questionnaire covered all DSM-IV axis disorders: depressive disorders, bipolar disorders, anxiety disorders (including panic attacks, generalized anxiety and obsessive-compulsive disorders), substance abuse (alcohol and drugs), social and public place phobia, post-traumatic stress disorders and eating disorders. As stated before, we have excluded patients with alcohol and drug addiction. In our cohort, there were no patients with alcohol and drug addiction, but 8 patients had a history of alcohol abuse. The alcohol abuse and dependence were defined using the DSM IV criteria. Recurrent depression is one of the two forms of depressive disorders in DSM-IV, while the other psychiatric disorders do not have a recurrent form in the DSM-IV classification. All other disorders except recurrent depression are considered to exist at the time of the interview if the patient fulfilled the criteria of the particular disorder in the SCID-I assessment. Results from SCID-I were pooled into a sum variable that described presence of any psychiatric disorder.

### 2.5. Return to Work Evaluation

The data on RTW were obtained utilizing electronically registered sick leave periods. These data are available to all the health professionals in Finland, being saved into patients’ medical records. Initially, data were collected retrospectively and evaluated with one-day accuracy, accounting for the individuals who returned to work part-time. The sick-leave length was thereafter verified by structured and pre-determined telephone interviews one year after the index MTBI. 

### 2.6. Statistical Analyses

We used IBM SPSS Statistics 24 (IBM Corp., Armonk, NY, USA) to perform the analyses. Skewed distributions were reported as medians with interquartile ranges (IQR) and normally distributed values in mean and standard deviation (SD). Skewed data were compared between groups using a non-parametric Mann–Whitney U test. Normally distributed data were compared using a *t*-test. Categorical variables were compared using a two-sided χ^2^ test. We assessed the association between time to RTW and the presence of a psychiatric disorder by survival curves and log-rank tests. Presence of any psychiatric disorder was used as a binary variable, and the significance level was assessed at the 0.05 level. The different psychiatric disorders were not analyzed separately vs. outcome (RTW, RPQ, GOSE).

## 3. Results 

The clinical characteristics of the patients (*n* = 103) are summarized in Table 2, while mechanisms of injury are shown in Table 3. In total, 96 patients underwent computed tomography imaging at presentation after MTBI, and all patients underwent MR imaging at a median of 10 days (IQR 7–12) after MTBI. We utilized the NINDS common data elements for traumatic brain injury while performing radiological evaluation [23]. Traumatic brain lesions were described in 32 patients (31.1%), including acute subdural hematoma in 19 patients (18.4%), epidural hematoma in 1 patient, traumatic subarachnoid hemorrhage in 17 patients (16.5%), traumatic intracerebral hemorrhage or hemorrhagic contusions in 18 patients (17.5%), and diffuse axonal injury in 25 patients (24.3%). Blood alcohol concentration (BAC) was verified for 26 patients (25.0%) at ED. At the moment of injury, 19 patients (18.5%) were verified to be under the influence of alcohol, and 4 patients were highly likely to be under the influence, according to patient records. The median BAC for these 26 patients was 1.6 ‰ (IQR 1.2–2.0).

### 3.1. Clinical Evaluation

The median RPQ score at one month was 13.0 (IQR 6.7–19.3) in patients with psychiatric disorders diagnosed at 3 months after MBTI compared to 8.5 (IQR 2.6–14.4) in those without one (*p* = 0.021). The median NOS-TBI was very close to zero in both groups. The median GOSE was 7.0 (IQR 7.0–8.0) in patients with a psychiatric disorder compared to 8.0 (IQR 7.0–8.0) in patients without one (*p* = 0.003). For current medication, 5 out of 103 patients were taking antidepressants, 3 patients were using benzodiazepines, and 4 patients were using other psychiatric medication (stabilizing, anxiolytic or antipsychotic). 

### 3.2. Psychiatric Evaluation

The median time from MTBI to psychiatric evaluation was 111 days (IQR 88–193), i.e., 3 months and 21 days. Psychiatric disorders were diagnosed in 26 patients (25.2%). The frequencies of disorders are shown in Figure 1. 

In our cohort, current depression was present in nine patients, of which four patients also had had a previous depressive episode; thus, they had recurrent depression. Furthermore, nine patients had a past single depressive episode, but were not currently depressed. Alcohol abuse was diagnosed in eight patients and social anxiety disorder in seven. At least two parallel psychiatric disorders were diagnosed in eight patients. No patient fulfilled the diagnostic criteria for PTSD. 

### 3.3. Return to Work

The status regarding return to work was acquired from 94 of the 103 patients: 18 out of 20 (90.0%) had a psychiatric disorder, and 76 out of 83 (91.6%) did not. Full-time students were included (11 out of 94 patients), and return to studies was considered as RTW. Of the remaining patients, four had retired before MTBI, and five patients were not employed for various reasons. While psychiatric evaluation was performed after 111 days median time, the RTW dynamics had longer follow up. The aim was to very accurately follow the RTW in all the subjects included to the study. At three months, there was no significant difference between patients with psychiatric disorders versus those without them in the rate of return to full-time work (95.7% vs. 87.3%, *p* = 0.260) or at least part-time work (100% vs. 94.4%, *p* = 0.245). In the Kaplan–Meier analysis (Figure 2), the median time to return to work (at least part-time) was 10.0 days for both groups (95% CI 3.1–16.9 days for patients with psychiatric disorders and 4.2–15.7 days for those without). Within one year, all but one patient without any psychiatric disorder had returned to work.

## 4. Discussion

In this prospective cohort study, we found that approximately every fourth patient with MTBI had a comorbid psychiatric disorder. The existence of psychiatric disorders did not affect the rate of return to work after MTBI. However, patients with psychiatric disorders reported more symptoms in RPQ than those without one, and the functional outcome measured with GOSE at one month after MTBI was worse in patients with psychiatric disorders compared to those without.

Overall, RTW after MTBI is usually high, that is, over 90% two months after MTBI as evaluated in two previous studies [24,25], which is in accordance with our results. As far as we know, there has previously been no such comprehensive analysis done on the influence of psychiatric disorders on RTW after MTBI. RPQ scores were higher in patients with psychiatric disorders. This is not surprising, as many items, e.g., depressive mood, sleep disorders, cognitive problems and fatigue, are also part of diagnostic criteria for depressive disorders or anxiety in the DSM IV. In those patients with high scores on the “psychiatric” items of the RPQ, it is sometimes not possible to distinguish between clinical depression and reactive symptoms after trauma. Post-concussive syndrome [26] has been suggested as an explanation for this kind of reactive symptom, but so far consensus about the syndrome has not been reached, since several factors behind the symptoms are conceivable. Concerning the RPQ score in our study being significantly higher and the GOSE score significantly lower in patients with psychiatric disorders, one would expect that RTW would have been delayed. However, this was not the case, and we speculate that early contact with healthcare professionals and continuous follow-up as part of the research protocol probably predisposed patients to favorable RTW as an unintentional intervention. 

Our result concerning functional outcome measurement with GOSE is in accordance with a recent study [27] in which patients with psychiatric history had worse outcomes after MTBI measured with GOSE than patients without it. Concerning RPQ score, we have not found a comparable analysis in patients with psychiatric disorders. The relationship between symptom reporting/functional outcome and psychiatric manifestations is probably bidirectional: first, suffering initial symptoms after TBI can evoke apprehensive feelings in patients and may trigger psychiatric morbidity. Second, a psychiatric condition (for example, depression) in the background can increase the severity of experienced symptoms. Our study does not comment on which mechanism is more central, but the absence of influence on RTW is apparent.

Depression was the most common mental disorder in our cohort, although it was less frequent than in comparable studies [28]. In a study of patients with moderate and severe TBI three months after incident trauma, the prevalence of depression (identified by the Hospital Anxiety and Depression Scale (HADS), -D score cut point) was clearly higher than in our study (56% vs. 26%), showing higher depression rates in patients with more marked traumatic changes [29]. Further, Bombardier et al. [30] used the Patient Health Questionnaire 9-item depression scale (PHQ-9) to diagnose depression at the one-year follow-up and identified depression in as many as 53% of their patients, finding no differences in depression rates after mild complicated, moderate and severe brain injury. The severity of injury with the presence of more neurological deficits and disabilities might secondarily affect the mental health of examined individuals and might cause a more remarkable presence of psychiatric pathology. On the other hand, using depression scales as diagnostic tools may result in higher estimates of depression than using structured clinical interviews (e.g., SCID).

The presence of anxiety disorders and PTSD after mild TBI has been previously reported with great variability among different studies [31]. According to the literature, anxiety disorder is more likely present after mild TBI compared to moderate and severe TBI, since amnesia and loss of consciousness appear to act as protective factors [32]. In our clinical cohort, 15 patients (14.5%) had anxiety disorder (generalized anxiety disorder, social phobia, agoraphobia, obsessive-compulsive disorder, panic disorder), which is in accordance with previous studies [3]. The coexistence of anxiety and depressive disorders is also commonly reported in MTBI patients [28], which was also found in three of our patients. We did not include specific mood or anxiety scales to measure the severity of the psychiatric symptoms, but we relied instead on SCID categorization.

In our study, no patients fulfilled the criteria for PTSD, which might be related to initial trauma mechanisms, with only four patients being the victims of interpersonal violence. The type of trauma essentially correlates with PTSD. The significant presence of PTSD pathology was described to be as high as 43.9% in war veterans after mild TBI [33], and it remains high in patients facing human-made, natural or technological disasters [34] or interpersonal violent events [35]. 

In war veterans, the PTSD and depression comorbidity after MTBI was shown to affect the dynamics of recovery in a recent study [36]. 

In our cohort, 36 patients had MRI-confirmed traumatic lesions, not exceeding mild complicated TBI criteria. The presence of intracranial lesions was not correlated to psychiatric findings. In a separate analysis of the same cohort previously reported by our group [37], intracranial lesions visible in initial CT showed a tendency to delay the early RTW during the first days or weeks after trauma, but they did not affect the outcome of GOSE. 

The association between post-concussion emerged psychiatric disorders and return to work is complicated. Firstly, from previous studies and clinical experience, it is well known that post-traumatic depression usually takes weeks to months to emerge. Identified as the most common psychiatric disorder in some patients in our cohort, depression is severe and inhibits RTW, but it is probably more common that the association is multi-faceted: depression is a surrogate marker of vulnerable features in personality or psychosocial stressors that affect performance. Thus, we suggest that there is an association, but not necessarily a causal relationship between the presence of psychiatric disorders at 3 months after MTBI and RTW. 

The strengths of this study are the precise evaluations of RTW and psychiatric disorders. RTW was evaluated by one-day accuracy. To assess psychiatric disorders, we used SCID, which is a reliable structured interview method compared to psychiatric parameter evaluations based on scales. SCID is a clinical tool with previously validated feasibility and excellent inter-rater reliability [38]. We assume that the threshold to objectively report mental health challenges has remained low, as the patients were aware that the study was conducted anonymously. Finnish as a native language was mandatory in order to establish the cohort largely sharing similar cultural values, as shown before [39], thuspsychiatric diagnosis was not likely culturally influenced or ethnicity-related [40]. 

A limitation of the study is that the previous psychiatric history of the patients was not reported in detail. Alcohol and/or drug addiction and previously diagnosed schizophrenia or schizoaffective disease were used as exclusion criteria, but other psychiatric conditions were not systemically screened. The reason why we did not have depression as exclusion criterium was that in previous literature prevalence of major depression has been reported to be high, 15–20% at 3–12 months after mTBI. If this had been the case in our study, it would have been necessary to analyze the possible (probable) effect of de novo post-TBI depression on RTW and differentiate the patients with recurrent depression from those with TBI-associating depression. Regarding potential previous psychiatric conditions, we have concentrated on previous depression because SCID has a systematic approach to this: previous depressive states must have been registered in order to reach a diagnosis of recurrent depression. From our data, we could not draw conclusions of the role of other previous psychiatric disorders in RTW. In a large epidemiological interview-based study [41], the one-year prevalence of depressive disorders in the working-age population of Finland has been around 6.5%. The prevalence of depression in our patients was somewhat higher. However, it is possible that some of the patients had undiagnosed depression or other undiagnosed mental illness before TBI, which might affect the results. While analyzing the results of our study, we accounted for the possible limitation of the theoretical possibility that patients who are motivated to become research study participants might carry more probability to return to work earlier than others being denied participation, which might represent self-selection bias. In addition, the cohort size is a limitation, leading to a modest frequency of psychiatric disorders, which does not allow an analysis of subgroups. We are also aware that some psychiatric disorders might be more debilitating than others, but detailed analysis on this was not possible due to the sample size. 

## 5. Conclusions

The presence of psychiatric comorbidity in MTBI was related to increased symptom reporting and worse functional outcome, even though it did not delay return to work. The results underline the importance of assessing psychological and psychiatric manifestations in patients with MTBI. In our study, as part of the research protocol, patients had early and recurrent contact with healthcare professionals and were followed up to one year. Early contact and adequate follow-up by healthcare professionals is important when supporting the patient’s return to work. An optimal, sustainable level of intervention should be defined, keeping the high incidence of MTBI in mind.

## Figures and Tables

**Figure 1 brainsci-10-00916-f001:**
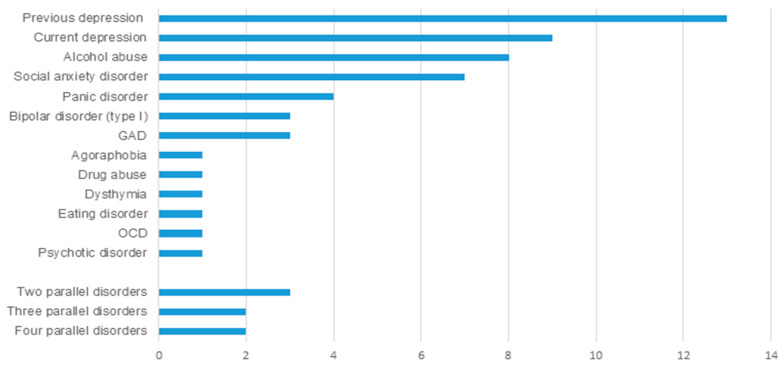
Frequency of psychiatric disorders. GAD: Generalized anxiety disorder; OCD: Obsessive-compulsive disorder.

**Figure 2 brainsci-10-00916-f002:**
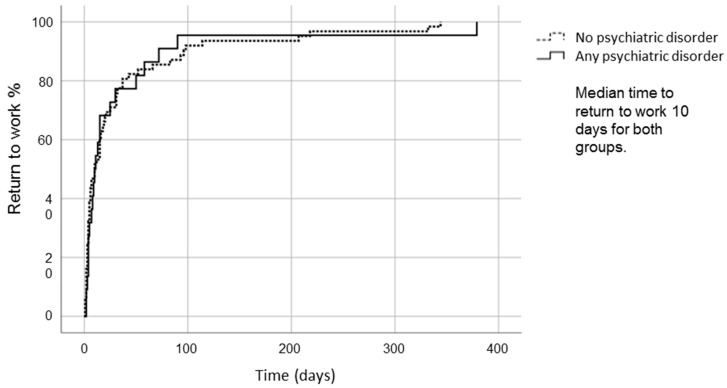
The effect of psychiatric disorder on return to work, as determined by Kaplan–Meier long rank analysis.

**Table 1 brainsci-10-00916-t001:** Comparison between patients dropped out from and included in the study.

	Dropped Out *n* = 28	Included *n* = 103	*p* Value
Mean age	37.5 (SD 12.5)	40.5 (SD 13.2)	0.265
Female sex	60.7%	42.7%	0.091
Median time to RTW	7.0 (4.7–9.3) days	10.0 (6.4–3.7) days	0.175
Median GOSE	8.0	8.0	0.378

Age: *t*-test. Gender: Pearson chi2 test. Time to return to work (RTW): Kaplan–Meier log rank analysis. Glasgow outcome scale extended (GOSE): Mann–Whitney U-test.

**Table 2 brainsci-10-00916-t002:** Characteristics of the study population.

Variable	No Psychiatric Disorder			Any Psychiatric Disorder			*p*
	Valid n	Mean/n	SD/IQR/%	Valid n	Mean/n	SD/IQR/%	
Mean age	77	41.6	13.0	26	38.5	3.2	0.293
Female sex	77	36	46.8	26	9	34.6	0.281
Married or cohabited	77	55	53.4	26	19	73.1	0.872
Education >15 years	77	47	61.0	26	12	45.6	0.185
GCS < 15 (= 13 or 14)	42	13	31.0	13	5	38.5	0.614
Median LOC hh:min	53	0:01	0:01–0:04	16	0:01	0:01–0:02	0.488
Median retrograde PTA hh:min	68	0:00	0:00–0:05	24	0:05	0:00–0:30	0.478
Median anterograde PTA hh:min	68	1:00	0:20–4:00	24	1:17	0:29–2:22	0.476
Traumatic lesion in MRI	77	26	33.8	26	10	38.5	0.418

Dichotomous variables: Pearson Chi2 test. Continuous variables: One-way Anova or Interquartile range. hh: min 0:00. GCS—Glasgow Coma Scale at emergency department. LOC—loss of consciousness. PTA—post-traumatic amnesia.

**Table 3 brainsci-10-00916-t003:** Mechanism of injury (*n* = 103).

Mechanism	N	%
Car accident	4	3.9
Motorcycle accident	2	1.9
Traffic accident as a pedestrian	2	1.9
Sports	13	12.6
Bicycle accident	28	27.2
Ground-level fall	28	27.2
Fall from a height	20	19.4
Violence	3	2.9
Other	2	1.9
Unknown	1	1.0

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
