# Peer review of "Prognosis after Mild Traumatic Brain Injury: Influence of Psychiatric Disorders"

_brainsci, 2020, doi:10.3390/brainsci10120916_

Reviewer 1 Report
Overall, this is an important manuscript. However, there are several issues that need to be considered.
Introduction:
- Lines 62-65, run-on sentence. Please break up into two
- The last paragraph of the introduction is all one sentence and does not read too well. Please break the run-on sentence for improved readability
- There seems to be a paragraph missing in the introduction. The transition from discussing psychiatric conditions and return to work (RTW) is not smooth at all. I suggest an introduction to RTW in the presence of mTBI. What is the current research on RTW following mTBI? What are the gaps in literature that this manuscript hopes to fill? Considering that RTW is a major focus of the study, more background information linking RTW with mTBI is warranted
Methods:
- Please explain why CT scans were completed or mTBI patients. mTBI can be easily diagnosed without a CT. It seems that a CT scan is almost a pre-requisite for study inclusion. Please explain the need for a CT scan
- Please justify the non-exclusion of depression. This study found that 4 patients had recurrent depression, which confounds the results—these patients may be more adept at navigating their condition, hence better able to RTW.
- Please include information regarding follow-up. The abstract indicates that patients were followed up for one year while psychiatric evaluation was after 111 days. Please clearly state the specific timepoints different measures were administered. Clarify the actual follow-up time and what was measured at follow-up.
- Please clarify why the hospital has RTW information. As the manuscript currently reads, it seems as if the hospital has RTW information. If this is actually true, please clarify this. If not, please include where this data is obtained from.
- Please define the measures used for clinical evaluation. What is included in the NOS-TBI and RPQ? A sentence or two regarding the components of these measures will help the readers understand the measures better
Results:
- Line 163 states 96 patients underwent a CT scan. Why is this number less than the 103 patients recruited for this study? Why did 7 not have a CT scan, when it is seemingly the inclusion criteria
- Digital RTW documentation is discussed here but it belongs in the methods. Furthermore, see comment above regarding RTW—what data source is this?
Discussion
- What was the significance between psychiatric conditions vs. non-psychiatric conditions? The authors state multiple times that one-fourth of the participants had a psychiatric conditions, but was a statistical test done to examine if this is statistically significant? A chi-square test would be sufficient and the results should be mentioned in the manuscript.
- On line 282, please add ‘likely’ to the last sentence and remove ‘guaranteeing’; the sentence worded as now is too definitive. It is highly likely the results are not culturally or ethnicity influenced, but not definitively.
- The authors should include a discussion on the likelihood of undiagnosed depression among the population. It is very possible that patients had depression prior to mTBI but were not diagnosed. Therefore, there is a possibility of inflated incidence and prevalence of depression following mTBI. Similarly, there is also a possibility of other undiagnosed psychiatric disorders, which then artificially inflates the proportion of participants with a psychiatric condition following TBI. Please include this in the discussion and perhaps in the limitation section.
- RPQ scores are higher for participants with a psychiatric condition and this is expected since RPQ has questions correlated with depression and anxiety. It is very important to include this in the discussion so readers can get a complete picture as to why these scores are higher. Similarly, it will be beneficial to include a discussion of post-concussive syndrome—it is possible that post-concussive syndrome symptoms are masked as a psychiatric condition
Overall
- Please go through manuscript for grammatical errors. There are certain places where words are missing.
- Please see above comment regarding run-on sentences.
Author Response
REVIEWER 1:
Introduction:
- Lines 62-65, run-on sentence. Please break up into two
Thank you for this important remark, we split the sentence according to the reviewer´s suggestion.
- The last paragraph of the introduction is all one sentence and does not read too well. Please break the run-on sentence for improved readability
Thank you for this valuable suggestion, we edited the last paragraph of introduction to make it more readable
- There seems to be a paragraph missing in the introduction. The transition from discussing psychiatric conditions and return to work (RTW) is not smooth at all. I suggest an introduction to RTW in the presence of mTBI. What is the current research on RTW following mTBI? What are the gaps in literature that this manuscript hopes to fill? Considering that RTW is a major focus of the study, more background information linking RTW with mTBI is warranted
Thank you for your constructive comment. We have now added additional references and additional paragraph considering RTW after MTBI (lines 71-73)
While MTBI clearly affects RTW (Iverson GL et al.), RTW after MTBI is mostly short (Bloom et al). However, the particular influence of psychiatric disorders in MTBI patients and their joint influence on RTW dynamics is so far understudied.
Methods:
- Please explain why CT scans were completed or mTBI patients. mTBI can be easily diagnosed without a CT. It seems that a CT scan is almost a pre-requisite for study inclusion. Please explain the need for a CT scan
In Finland, it is a routine to do the CT scan as low threshold examination to most of the head trauma patients being examined in emergency, including those with clinically present mTBI, to exclude the theoretical possibility of more severe trauma, and as well providing the reassurance to the patient about the benign nature of disease.
Now, we have added the following sentence into Methods (2.3 Brain Imaging) lines 135-136
”Most of the patients (96 out of 103) underwent brain CT scan in emergency, following a national consensus recommendation which recommends CT scan when certain criteria are filled.”
- Please justify the non-exclusion of depression. This study found that 4 patients had recurrent depression, which confounds the results—these patients may be more adept at navigating their condition, hence better able to RTW.
When designing our study, we expected the prevalence of post-TBI depression to be much higher than what we actually found. In the previous literature, prevalence of major depression has been reported to be 15-20% at 3-12 months after mTBI. Had that been the case in our study, it would have been necessary to analyze the possible (probable) effect of de novo post-TBI depression on RTW and differentiate the patients with recurrent depression from those with TBI-associated depression.
Related to this, we have now added the following section to the discussion (lines 323-327)
”The reason why we did not have depression as exclusion criterium was that in previous literature prevalence of major depression has been reported to be high, 15-20% at 3-12 months after mTBI. If this had been the case in our study, it would have been necessary to analyze the possible (probable) effect of de novo post-TBI depression on RTW and differentiate the patients with recurrent depression from those with TBI-associating depression.”
- Please include information regarding follow-up. The abstract indicates that patients were followed up for one year while psychiatric evaluation was after 111 days. Please clearly state the specific timepoints different measures were administered. Clarify the actual follow-up time and what was measured at follow-up.
While psychiatric evaluation was performed after 111 days median time, the RTW dynamics had longer follow up. The aim was to very accurately follow the RTW in all the subjects included to the study.
We have now added this sentence to the Results (3.3) lines 228-230
- Please clarify why the hospital has RTW information. As the manuscript currently reads, it seems as if the hospital has RTW information. If this is actually true, please clarify this. If not, please include where this data is obtained from.
Thank you for this inquiry. In Finland, the sick leave is registered always electronically. The data on RTW was obtained utilizing the electronically registered sick leaves. This data is available to all the health professionals in Finland, being saved into patients medical records
The clarification of this was added into Methods section 2.5 , lines 166-168
- Please define the measures used for clinical evaluation. What is included in the NOS-TBI and RPQ? A sentence or two regarding the components of these measures will help the readers understand the measures better
Thank you for your important suggestion. We have now added the following sentences to Methods, undersection 2.2. Clinical Evaluation, lines 121-127
The overall neurological condition was evaluated using the Neurological Outcome Scale for TBI (NOS-TBI), a measure developed to identify essential neurological deficits affecting the individuals with TBI [18]. Additionally the Rivermead Post-Concussion Symptoms Questionnaire (RPQ) as self-report tool to assess the severity of post-consussion symptoms [19], and the Extended Glasgow Outcome Scale (GOSE) [20] as a global outcome measure in TBI survivors were utilized for analyses of clinical symptoms.
Results:
Line 163 states 96 patients underwent a CT scan. Why is this number less than the 103 patients recruited for this study? Why did 7 not have a CT scan, when it is seemingly the inclusion criteria
CT scan was not an inclusion criterium, but as we stated above:
”Most of our patients (96 out of 103) underwent brain CT scan in emergency, following a national consensus recommendation which recommends CT scan when certain criteria are filled.”
This sentence was added to Methods (2.3 Brain Imaging) , lines 135-136
- Digital RTW documentation is discussed here but it belongs in the methods. Furthermore, see comment above regarding RTW—what data source is this?
Thank you for this remark, we have now removed this sentence. The method section related to this was edited.
Discussion
- What was the significance between psychiatric conditions vs. non-psychiatric conditions? The authors state multiple times that one-fourth of the participants had a psychiatric conditions, but was a statistical test done to examine if this is statistically significant? A chi-square test would be sufficient and the results should be mentioned in the manuscript.
Thank you for this focused comment. As we had no control group in this analysis, we did not conduct a statistical test on this. Instead we would like to point out that we have added text addressing the general prevalence of depressive disorders in Finland, see below.
- On line 282, please add ‘likely’ to the last sentence and remove ‘guaranteeing’; the sentence worded as now is too definitive. It is highly likely the results are not culturally or ethnicity influenced, but not definitively.
Thank you for pointing out this. We have changed the word guaranteeing by the word likely (line 318-319)
- The authors should include a discussion on the likelihood of undiagnosed depression among the population. It is very possible that patients had depression prior to mTBI but were not diagnosed. Therefore, there is a possibility of inflated incidence and prevalence of depression following mTBI. Similarly, there is also a possibility of other undiagnosed psychiatric disorders, which then artificially inflates the proportion of participants with a psychiatric condition following TBI. Please include this in the discussion and perhaps in the limitation section.
Thank you for your constructive comment. We now added the following paragraph to the discussion (lines 331-335):
In a large epidemiological interview-based study (Pirkola S, et al.), the one-year prevalence of depressive disorders in the working-age population of Finland the prevalence of depressive disorders has been around 6.5%. The prevalence of depression in our patients was somewhat higher. However, it is possible that some of the patients had undiagnosed depression or other undiagnosed mental illness before TBI, which might affect the results.
- RPQ scores are higher for participants with a psychiatric condition and this is expected since RPQ has questions correlated with depression and anxiety. It is very important to include this in the discussion so readers can get a complete picture as to why these scores are higher. Similarly, it will be beneficial to include a discussion of post-concussive syndrome—it is possible that post-concussive syndrome symptoms are masked as a psychiatric condition
Thank you for suggesting this important correction. We have now added the additional text to discussion (lines 247-254)
“RPQ scores were higher in patients with psychiatric disorders. This is not surprising, as many items, eg. depressive mood, sleep disorders, cognitive problems and fatigue are also part of diagnostic criteria for depressive disorders or anxiety in the DSM IV. In those patients with high scores on the “psychiatric” items of the RPQ it is sometimes not possible to distinguish between clinical depression and reactive symptoms after trauma. Post-concussive syndrome (Bigler E et al..) has been suggested as an explanation for this kind of reactive symptoms, but so far consensus about the syndrome has not been reached, since several factors behind the symptoms are conceivable
Overall
- Please go through manuscript for grammatical errors. There are certain places where words are missing.
Thank you for this remark, we have now performed the thorough English language check- up, by language professional, and the language editing was performed throughout the manuscript.
- Please see above comment regarding run-on sentences
We also tried our best to make the changes related to this issue accordingly.
Reviewer 2 Report
Dear Editor
The prospective cohort study by Marinkovic et al. reports that approximately every fourth patient with MTBI had a comorbid psychiatric disorder. The existence of psychiatric disorder did not affect the rate of return to work after MTBI. However, patients with psychiatric disorder reported more symptoms in RPQ than those without it, and the functional outcome measured with GOSE at one month after MTBI was worse in patients with psychiatric disorder compared to those without. The results point out to the importance of assessing psychological and psychiatric manifestations in patients with MTBI.
The design of the study and the technical quality of the work are convincing and results can be of general interest. The manuscript is well-written and easy to follow. Authors did well in mentioning the limitations of their study and have successfully managed to discuss the major findings of their study through an unbiased comparison with literature.
However, there is a number of major and minor points that would need to be addressed in order to improve the quality of this paper before it can be accepted for publication:
Major:
Authors need to perform a post hoc correction to moderate the variability in comparing multiple samples. Bonferroni correction or Conover-Inman post hoc corrections must be performed. Authors need to provide the new corrected p values and report any difference in their observations.
Minor:
-Line 98: authors need to mention how they define both alcohol abuse and dependence.
-Line 238-239: et al. instead of “coauthors”.
Best
Author Response
REVIEWER 2
-Authors need to perform a post hoc correction to moderate the variability in comparing multiple samples. Bonferroni correction or Conover-Inman post hoc corrections must be performed. Authors need to provide the new corrected p values and report any difference in their observations.
Thank you for this focused comment. Presence of any psychiatric disorder was used as a binary variable, and the different psychiatric disorders were not analyzed separately vs outcome (RTW, RPQ, GOSE). Therefore significance level was assessed at the 0.05 level. We have now clarified this use of binary variable with the following changes:
1) In Methods , subsection 2.6 , we have now added the following explanation (lines 179-181)
”Presence of any psychiatric disorder was used as a binary variable, and the significance level was assessed at the 0.05 level. The different psychiatric disorders were not analyzed separately vs outcome (RTW, RPQ, GOSE).”
2) In Methods , subsection 2.4 we have now added the following (lines 163-164)
”Results from SCID-I were pooled into a sum variable that described presence of any psychiatric disorder.”
3) In abstract, as additional clarification we added “presence of any” and changed the sentence accordingly (line 50)
However, presence of any psychiatric disorder did not affect RTW
-Line 98: authors need to mention how they define both alcohol abuse and dependence.
Thank you for your suggestion. We now specified this by adding the following sentence in Methods , subsection 2.4 Psychiatric evaluation (line 159)
“The alcohol abuse and dependence were defined using the DSM IV criteria. “
-Line 238-239: et al. instead of “coauthors”.
Thank you, we changed “coauthors” by “et al.” (line 273 )
Round 2
Reviewer 1 Report
Thank you very much for the very comprehensive revisions. This is an important manuscript that warrants publication. One final comment:
In the introduction, can you please add "following TBI" in the first sentence of the second paragraph of the introduction (line 66). The sentence should read:
"Return to Work (RTW) following TBI is a complex outcome parameter influenced by numerous factors, such 66 as physical, cognitive, and emotional challenges."
Reviewer 2 Report
Dear editor
I would like to thank the authors for their efforts to revise the manuscript in the light of the raised concerns and suggestions. All my comments have been addressed by the authors accordingly.
I would like to recommend this manuscript for publication.
Best